# The Influence of Short-Term Kinesiology Taping on Foot Anthropometry and Pain in Patients Suffering from Hallux Valgus

**DOI:** 10.3390/medicina57040313

**Published:** 2021-03-26

**Authors:** Tobiasz Żłobiński, Anna Stolecka-Warzecha, Magdalena Hartman-Petrycka, Barbara Błońska-Fajfrowska

**Affiliations:** Department of Basic Biomedical Science, Faculty of Pharmaceutical Sciences in Sosnowiec, Medical University of Silesia in Katowice, 41-205 Katowice, Poland; astolecka@sum.edu.pl (A.S.-W.); mhartman@sum.edu.pl (M.H.-P.); bbf@sum.edu.pl (B.B.-F.)

**Keywords:** hallux valgus, foot, kinesiology taping, anthropometry of foot, pain

## Abstract

*Background and Objectives*: Hallux valgus, one of the most common foot disorders, contributes to the formation of pain, changes foot proportions and hinders everyday functioning. In this study we wanted to verify if kinesiology taping improves hallux valgus and affects the position as well as reducing pain. *Materials and Methods:* Forty feet with hallux valgus were examined and the parameters were measured at three stages: before the kinesiology taping was applied, just after its application and after a month of use. Measurements were taken with a 3D scanner and a baropodometric platform. *Results:* When taping was applied, the hallux valgus angle decreased statistically significantly compared with pre-taping (*p* < 0.01). The use of taping for a month significantly reduced this angle compared with pre-taping (*p* < 0.05). Parameters such as foot length, the surface of the hindfoot and forefoot and hindfoot pressure on the ground changed. A change in the hallux position due to the taping produced significant changes in the perception of pain (*p* < 0.001). *Conclusion:* Kinesiology taping acts on the hallux valgus and foot position mechanically. This makes kinesiology taping an effective method of conservative treatment for patients who are not qualified for surgery.

## 1. Introduction

Hallux valgus (HV) is the most common osteoarticular deformation occurring within the forefoot [1,2]. The authors of a study conducted in Turkey with 2662 adult participants estimated the frequency of HV at 54.3% [1] and predisposing factors include the female gender, family history, improper footwear and age [1,2]. Flat feet and hindfoot valgus [3] as well as an imbalance in muscle tone [4] are also considered to be possible causes of HV but in most cases it is regarded as a multifactorial condition [1,3,4].

HV may occur in varying degrees from minor asymptomatic changes to significant deformations that disrupt everyday functioning. The most frequently reported problems by patients are pain caused by degenerative changes in the first metatarsophalangeal (MTP) joint, metatarsalgia [5], skin changes (i.e., calluses, corns, ingrown nails) [6] and painful overloading of soft tissues in the foot as well as pain in the knee, hip or spine [2]. Patients very often also report a problem with selecting footwear due to the changes in the proportion and appearance of the foot, which can lead to a reduction in self-esteem and psychological disorders [2,4].

The treatment of HV includes conservative management such as exercises [4,7], manual therapy [8], physical therapy [8,9], orthoses [10], insoles [11], kinesiology taping [4,12] and surgical treatment [13]. Surgical treatment includes many different techniques such as a Scarf osteotomy, Chevron osteotomy, Silver procedure, Lapidus procedure and Akin osteotomy [13].

Kinesiology taping is a therapeutic method that uses the mechanical effect of a piece of elastic adhesive tape on the musculoskeletal system [14]. Kinesiology taping uses cotton with elastic filament tapes covered with acrylic adhesive on one side. The advantage of the tape is its continual effect 24/7 and the fact that the application remains on the skin for several days. Each tape application should be preceded by an examination of the patient and an assessment of the musculoskeletal system [14]. The kinesiology taping technique can be helpful in hip [15], knee [16] and foot instability [17] or HV deformity [4,12] pain syndromes.

There are not many reports to be found that describe the results of studies about the impact of kinesiology taping [18,19] and non-elastic taping [4,12,20,21] on HV and these studies do not include linear measurements of the foot and load measurements of individual parts of the foot in a standing/static position.

The aim of this study was to analyze the impact of using kinesiology taping on the anthropometric characteristics of the foot as well as on the sensation of pain in the foot with the HV deformity. This aim is part of the search for an effective conservative treatment that will help patients who are not qualified for surgery for various reasons.

We evaluated the objective parameters immediately after first using the kinesiology taping and after its extended use.

Considering that not every patient can or wants to undergo surgery, effective methods of conservative treatment should be sought to help such patients.

## 2. Materials and Methods

The research project was consistent with the Declaration of Helsinki and was approved by the Bioethical Committee of Medical University of Silesia (approval number: KNW/0022/KB1/27/I/16). Before starting the study, all participants were clearly informed about the purpose and method of the research and their involvement in the entire process. All participants signed a consent form to participate in the experiment. In accordance with the recommendations and guidelines of the Bioethical Committee of Medical University of Silesia, the form contained information on the objectives of the intended research and the manner of conducting it.

Twenty-three people aged 26.3 to 78.4 years, mean ± standard deviation: 55.8 ± 16.3 years, were involved in the study. They were all patients with symptomatic HV who attended the Healthy Foot Clinic and wanted to undergo a non-operative treatment and expressed their willingness to participate in the study. All of them signed their consent to participate in the study. Forty feet were examined including those of two males. The patient examinations were preceded by an interview to exclude people with contraindications to treatment using kinesiology taping. A physical examination was then carried out including a visual assessment of the feet, a manual examination of mobility within the joints of the feet, provocation tests and, where necessary, orthopedic and/or neurological tests. These were undertaken with the purpose of excluding contraindications (e.g., skin changes or wounds at the site of the potential course of taping, sensory disturbances, pain in the MTP joint I while moving, a positive Mulder test for interdigital neuroma) to undertake specific therapeutic actions.

Measurements of the foot pressure on the ground (PoG) aimed at assessing the plantar surface of the foot were made with a 3D sub scanner (PODOSCAN3D—3D laser foot scanner, Sensor Medica, Italy) (Figure 1a,b). This device works based on the well-developed podoscopic method. Its software performs angular and axial measurements of the foot including the length and width of the foot (mm), length of the longitudinal arch (mm) and the angle of the HV (°). In addition to the exact imprint (plantocontourogram), information is obtained about the spatial shape of the arch. This allows a reliable assessment of the anthropometric condition of the foot. 

Static foot PoG was measured using a baropodometric test performed on the FreeMED MAXI Platform (Sensor Medica, Italy) with a length of 260 cm of which the central active part is 50 cm × 60 cm (Figure 1c,d) and the end passive sections are 100 cm (sampling frequency up to 250–400 Hz real-time, square-shaped resistance sensors, 24 carat gold plated). This device tests the distribution of foot pressure on the ground in a standing position. The assessment of the loading on individual parts of the foot includes parameters such as the forefoot and hindfoot surface (cm^2^) and forefoot and hindfoot PoG (%). PoG values were calculated automatically (a) for a given part of each foot, assuming that both feet = 100% (PoGb) and (b) for the forefoot of each taped foot, assuming that a single treated foot = 100% (PoGt).

Measurements of foot parameters were made three times: before using the taping (T0), directly after applying the taping (T1) and after one month of taping use after the taping had been removed (T2).

After taking measurements at T1 and T2, the patients were asked to mark on the 10 cm linear scales (Figure 1e) their subjective feelings regarding changes in pain symptoms. Placing a mark between 0 and −5 meant deterioration while a mark between 0 and +5 meant improvement. The results are presented in the form of numbers in the range from −5 to +5.

Mueller Tuffner Pre-Tape adhesive and 5 cm wide kinesiology taping (3NS TEX) were used on the feet of patients. The toes were taped with taping of a prescribed length cut into the shape of the letter Y (Figure 2b) with the tails wrapped around the proximal phalanx of the toe (without stretching the tape) and the base applied along the medial edge of the foot (Figure 2e) up to the tuber calcanei with maximum tape tension (mechanical correction 75–100%). A second I-shaped strip (Figure 2c) was then applied from the medial edge of the first metatarsal bone to the lateral edge of the fifth metatarsal bone with maximum tape stretch (mechanical correction 75–100%). It was meant to improve the shape of the transverse arch of the foot and to stabilize the first metatarsal bone in the cuneiform joint (Figure 2d). Particular attention was paid so that this tape did not go past the lines of the MTP1 and MTP5 joints, which could cause adduction and pinching of the toes. The tails (the end parts of the tape), however, were attached without stretching on the dorsal surface of the foot and obliquely towards the tarsal bones (Figure 2f). The tape stayed on the foot for several days (depending on the sweating of the skin), which is why all patients were taped with a kinesiology taping application by the same physiotherapist every 2–3 days in a clinic in a period of about a month (28.1 ± 7.7 days on average). After this time patients came for measurements with the taping on but took it off just before the T2 examination.

The statistical analysis was performed using the Statistica 13.0 program. To assess the normality of the result distribution, the Shapiro–Wilk test was used and to assess the homogeneity of variance, Levene’s test was used. Due to the fact that the distribution of results for the subjective feelings of the patients differed from the normal distribution the Wilcoxon matched pairs test was used in the calculations. The distribution of the results for a few of the measurements of foot parameters also differed from the normal distribution, which is why the Friedman ANOVA test and Dunnett’s post hoc test were used to analyze the impact of taping on the foot static. The significance level of values *p* < 0.05 were considered statistically significant.

## 3. Results

Taping had a positive effect on the subjective symptoms of the patients. None of the patients reported that there was an increase in pain just after the application of the tapes (T1) or after one month of use (T2) (Figure 3). Moreover, at both time points, most of the patients declared a reduction in pain. This reduction in pain was significantly greater after a month of taping than just after the application of the taping (*p* < 0.001). The median reduction of pain immediately after the application of the taping was 1.9 and after a month of use it was 2.4.

The taping affected the HV angle (*p* < 0.001) (Figure 4). After applying the taping (T1), the angle decreased in a statistically significant way compared with the situation before the taping (T0) (*p* T0 vs. T1 < 0.01). The reduction in the HV angle affected all of the tested feet. The median (Me) and first (Q1) and third (Q3) quartile of this parameter at the T0 examination were Me: 17.0°; Q1: 13.5°; Q3: 21.5° and at the T1 examination Me: 10.5°; Q1: 6.0°; Q3: 14.5°. Taping for a month (T2) had a significant effect on the reduction of the HV angle compared with pre-taping (*p* T0 vs. T2 < 0.05). In 30% of the tested feet, the HV angle immediately after removing the taping (T2) was similar to the angle before the start of the treatment (T0) despite taping having been used for the month before. In 70% of the tested feet the value of this parameter remained lower than before the treatment. The median HV (Me) and the first (Q1) and third (Q3) quartile of this parameter at T3 were Me: 15.0°; Q1: 12.0°; Q3: 20.5°.

Taping affected the length of the foot (*p* < 0.001) (Table 1). After the application of the taping, the length of the foot increased significantly (*p* T0 vs. T1 < 0.01) and after one month of taping the foot length was still greater than before the kinesiology taping application (*p* T0 vs. T2 < 0.05). The width of the foot and the length of the longitudinal arch of the foot did not undergo any significant changes (Table 1). The taping did not cause any statistically significant changes in the forefoot area; however, taping increased the hindfoot area (*p* < 0.05) (Table 1). A statistically significant increase in the surface occurred immediately after the application of the taping (*p* T0 vs. T1 < 0.05) but when this was removed after a month, this effect decreased and was not statistically significant. Forefoot PoGb after the application of the taping was reduced (*p* < 0.01) with this effect being statistically significant just after the taping was applied (*p* T0 vs. T1 < 0.05); however, following the tape’s removal after a month, it decreased so that it ceased to be statistically significant (Table 1). The reverse effect was observed in the hindfoot (*p* < 0.01): hindfoot PoGb increased after the application of the taping (*p* T0 vs. T1 < 0.05) and then decreased so that it ceased to be statistically significant. Forefoot PoGt indicated that taping had a significant effect on the changes in foot loading distribution (*p* < 0.001). Shortly after the taping was applied, the value of this (*p* T0 vs. T1 < 0.01) and this effect was maintained when the taping was removed after a month of use (*p* T0 vs. T2 < 0.05).

## 4. Discussion

Hallux valgus, one of the most common foot problems, can range from minor asymptomatic changes to severe deformities of the foot. These deformities cause pain, which seriously interferes with the everyday functioning of the patient and reduces their enjoyment of life [3,22]. In addition, a consequential change in the appearance of the foot is perceived by patients as a significant aesthetic defect, which like all defects of this type can lead to a decrease in self-esteem and psychological disorders. Considering that not every patient can or wants to undergo surgery, effective methods of conservative treatment should be sought to help such patients. The promising results obtained in our study showed that using kinesiology taping might be the method of conservative treatment.

Research by Nakagawa et al. [23] showed that despite the HV angle not changing during the study, the long-term use of orthoses as a non-surgical method of treating HV reduces pain and improves the quality of life of patients. However, a reduction of the HV angle was found in most studies where non-elastic taping was used instead of an orthoses [4,12,20,21].

Lee and Lee [18] and Jeon et al. [19] used kinesiology taping. The former presented a case study in which the HV angle in a patient was reduced from 21° to 14° while the latter, who examined 23 feet in 15 patients, improved the HV angle from an average of 21.95° to 18.74°.

The results presented in this paper are consistent with the results cited above. The HV angle improved immediately after applying the kinesiology taping; the median value of the HV angle decreased in all patients by an average of 6.5°. Compared with the studies of other authors, the method of measuring the HV angle was an important difference. In this study, 3D foot scans were used while in previous studies measurements were taken using a goniometer.

Many studies have shown the positive effect of kinesiology taping on the musculoskeletal system [15,17,24,25,26]. Yen et al. [17] indicated an improvement in the functioning of the joints of the lower limb and a resulting improvement in the stability of the ankle. Various authors have shown that kinesiology taping can be used to combat pain symptoms associated with musculoskeletal injuries [15,26,27]. Kinesiology taping supports injury prevention [28] and improves lymphatic flow [29].

In our study, the benefits of kinesiology taping translated into a significant improvement in the condition of the patient suffering from pain and local inflammation associated with HV. Patients experienced a significant reduction in pain symptoms during the month’s treatment with taping as well as after its removal. This confirmed the studies of other authors [18,19] including those who used non-elastic taping [4,12,21].

It may be assumed that wearing kinesiology tape changed the HV angle and, because of the temporary correction in the position of the big toe and the reduction in the pathological stretch of the ligaments and joint capsule on the medial side of the MTP joint, this subsequently diminished the pain in this area of the foot. The reduction in pain is reason enough to recommend kinesiology taping for use in the symptomatic treatment of this foot deformity.

There is no evidence in the current literature indicating changes in the anthropometric parameters of the foot after the application of the taping. This study revealed the effect that kinesiology taping had on the change in foot length in patients with HV. There was, however, no significant change in foot width as a result of using kinesiology taping. Most patients complain about problems selecting footwear because of the increase in the width of the foot due to the deformed position of the first metatarsal bone.

There are also no studies that show the impact of using kinesiology taping on the pressure of the different foot parts on the ground. Our research clearly showed that taped feet had a decrease in forefoot PoGb and an increase in hindfoot PoGb along with an increase in hindfoot surface. This effect seemed to be maintained in the treated foot, which resulted in a change in the loading ratio and a shift in the center of gravity towards the heel. This may be an important factor in reducing pain symptoms especially in the case of metatarsalgia co-existing with HV due to the lowering of the transverse arch of the foot and associated calluses [30].

The change in the HV angle using the kinesiology taping technique can be helpful in the treatment of conditions co-existing with HV deformity such as an ingrown nail on the lateral side of the big toe, corns between the toes or, in extreme cases, wounds caused by excessive pressure from the big toe on the second toe [6,30].

The limitations of our study included a small research group; in the study we focused on the assessment of changes in the anthropometric parameters of the foot, hallux valgus angle and parameters of the static load of the foot. An additional aspect was also the assessment of subjective pain sensation in patients, which could be extended with the use of more complex forms and pain assessment scales for the foot. However, the promising results obtained in the studies showed the need for further research on a larger group of patients and longer follow-up in this area to assess the long-term effects of kinesiology taping in patients with a hallux valgus deformity.

## 5. Conclusions

Kinesiology taping is a technique that is able to effectively reduce pain symptoms and change the position of the big toe in patients suffering from HV. The application of the taping on the big toe proves to the patient that the correction in the position of the big toe has a positive effect on the everyday functioning of the foot and on the reduction of pain. Therefore, those patients who for some reason do not want or cannot undergo HV corrective surgery can benefit from this conservative treatment option and thus improve the quality of their lives.

## Figures and Tables

**Figure 1 medicina-57-00313-f001:**
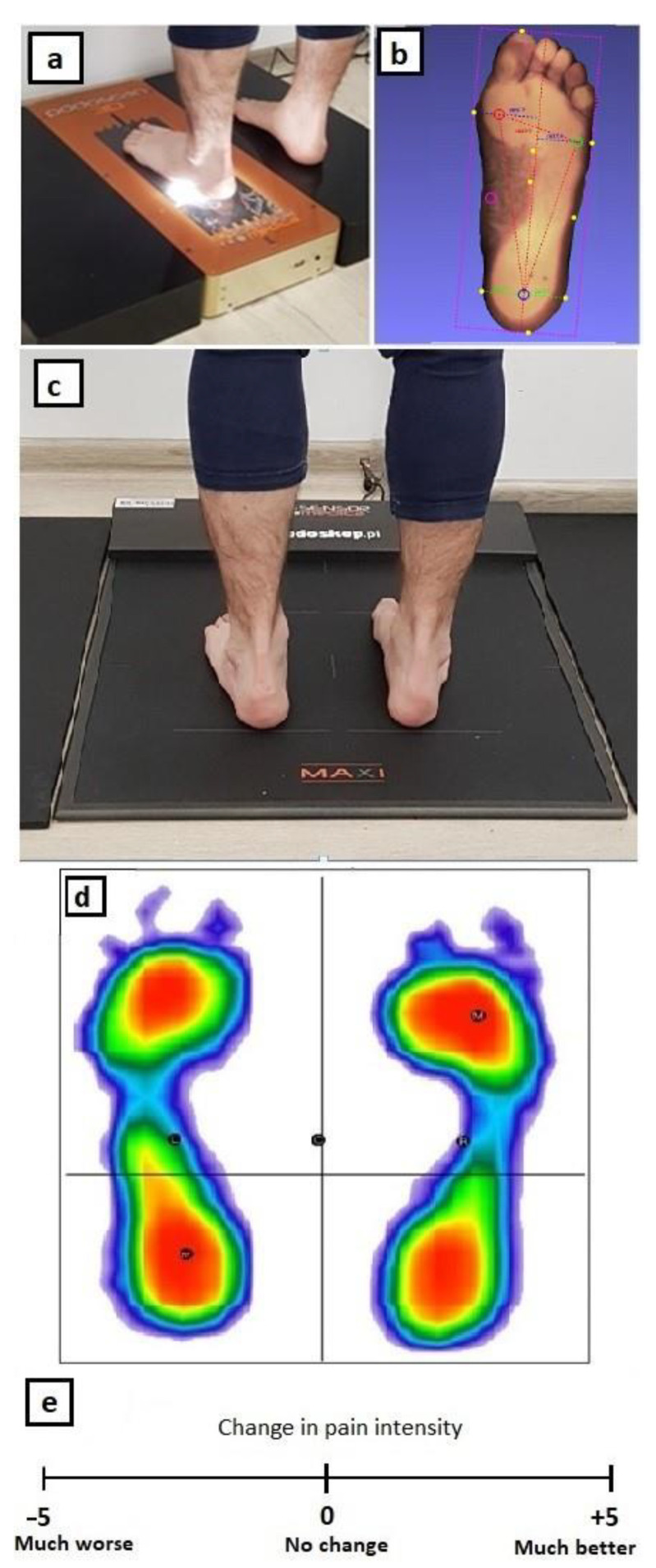
Performing a 3D scan of the left foot (**a**) and an example result (**b**), performing a static baropodometric test (**c**) and an example result (**d**), linear scales of subjective feelings regarding a change in functionality and pain symptoms of the foot where the distance (mm) from the “0” point with the appropriate sign constituted the numerical response of the patient (**e**).

**Figure 2 medicina-57-00313-f002:**
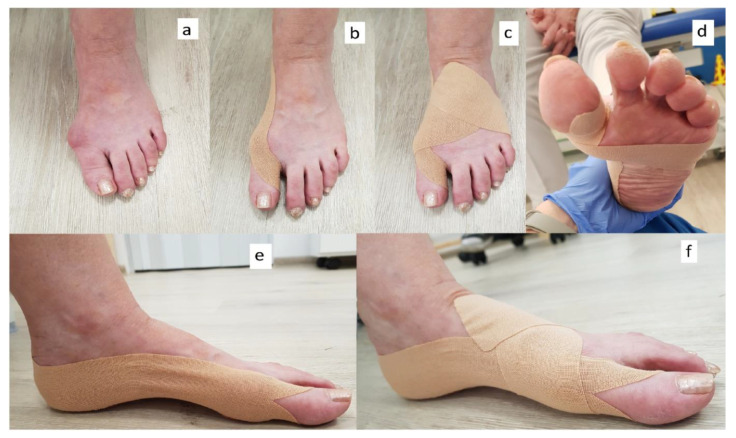
Application of the kinesiology taping. Foot before application: view of the dorsal surface (**a**), foot after the application of the first Y-shaped tape: dorsal (**b**) and medial (**e**) side, foot after the application of the second I-shaped patch: dorsal (**c**), plantar (**d**) and medial (**f**) side of the foot.

**Figure 3 medicina-57-00313-f003:**
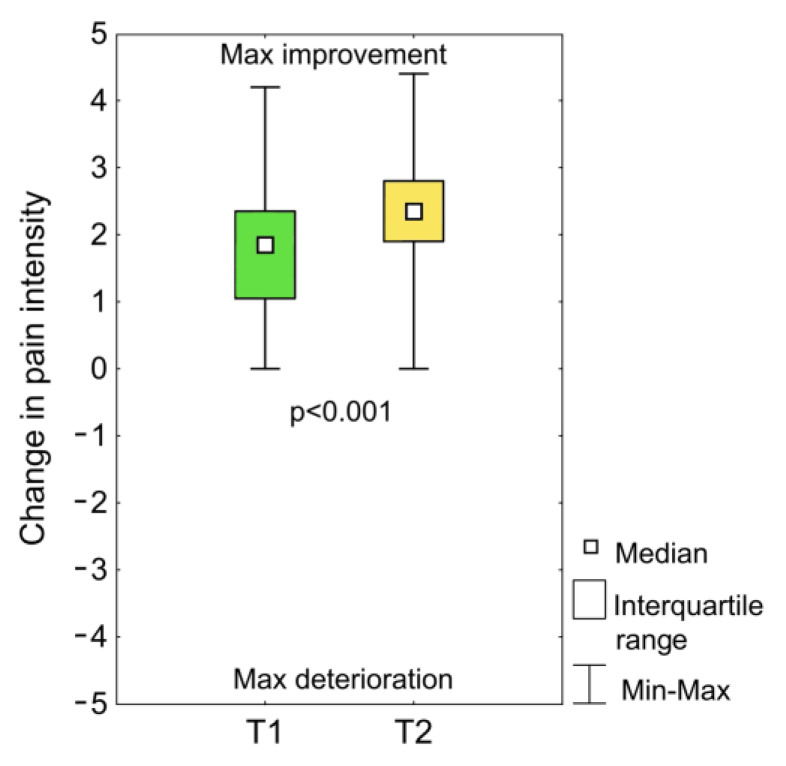
Change in pain intensity after the initial taping (T1) and after one month of use after the removal of the tape (T2); *p* = statistical significance, 0 = no change, 1 to 5 = improvement or decrease in pain, −1 to −5 = deterioration or increase in pain.

**Figure 4 medicina-57-00313-f004:**
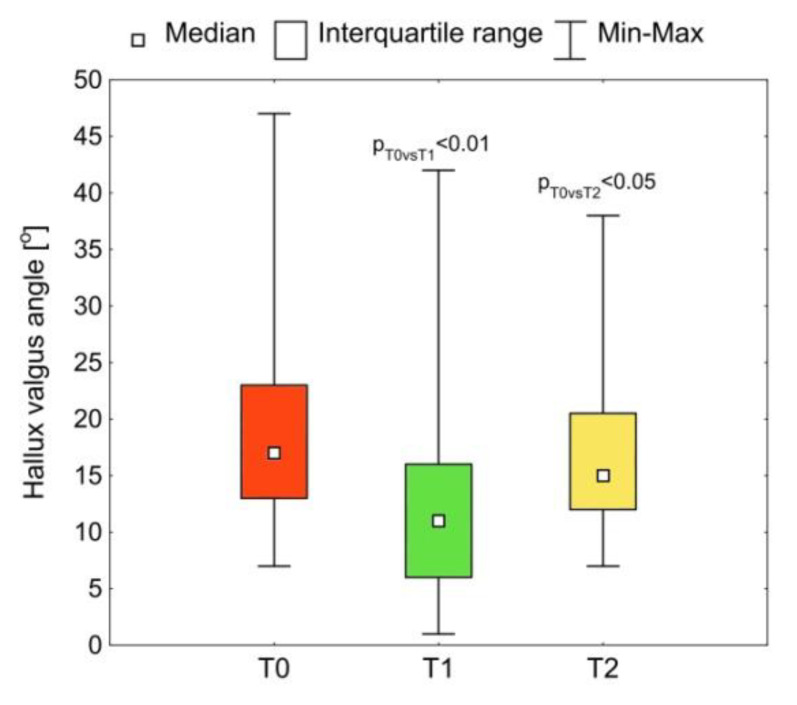
The hallux valgus (HV) angle before taping (T0), immediately after taping (T1) and after the removal of the tapes after a month of use (T2); *p* = statistical significance.

**Table 1 medicina-57-00313-t001:** Selected anthropometric and static foot parameters before using the kinesiology taping (T0), after the application of the taping (T1) and after its removal following a month of using the taping (T2); Q1 = first quartile, Q3 = third quartile, Min = minimum, Max = maximum, *p* = statistical significance, ns = no statistical significance.

Parameter	Time	Median	Q1	Q3	Min	Max	*p*	*p* T0 vs. T1	*p* T0 vs. T2
Foot length (mm)	T0	244.0	236.5	253.0	224.0	273.0	<0.001	-	-
	T1	245.0	238.5	253.5	224.0	276.0		<0.01	-
	T2	245.0	237.5	253.5	223.0	275.0		-	<0.05
Foot width (mm)	T0	93.0	90.0	97.5	82.0	105.0	ns	-	-
	T1	93.0	89.0	96.5	83.0	106.0		-	-
	T2	93.5	89.5	97.5	83.0	103.0		-	-
Length of the longitudinal arch of the foot (mm)	T0	176.5	171.0	183.5	160.0	199.0	ns	-	-
	T1	175.0	170.0	182.5	161.0	198.0		-	-
	T2	176.0	172.0	184.0	162.0	196.0		-	-
Surface of the forefoot (cm^2^)	T0	84.00	75.50	91.00	50.00	111.00	ns	-	-
	T1	78.00	68.50	88.50	52.00	121.00		-	-
	T2	82.00	67.00	94.00	47.00	111.00		-	-
Surface of the hindfoot (cm^2^)	T0	58.00	51.50	65.00	39.00	80.00	<0.05	-	-
	T1	60.00	54.50	66.00	40.00	86.00		<0.05	-
	T2	61.00	52.50	67.00	35.00	84.00		-	ns
Forefoot PoGb *(%)	T0	28.00	25.50	29.00	22.00	34.00	<0.01	-	-
	T1	26.00	24.00	27.00	21.00	31.00		<0.05	-
	T2	26.00	24.00	28.00	21.00	32.00		-	ns
Hindfoot PoGb *(%)	T0	22.50	21.00	25.00	17.00	29.00	<0.01	-	-
	T1	25.00	23.00	26.00	19.00	31.00		<0.05	-
	T2	24.00	21.00	26.00	17.00	32.00		-	ns
Forefoot PoGt **(%)	T0	55.00	51.50	59.00	46.00	65.00	<0.001	-	-
	T1	52.00	50.00	53.50	41.00	61.00		<0.01	-
	T2	52.00	49.00	56.00	45.00	62.00		-	<0.05

* Pressure on the ground value calculated automatically for a given part of each foot assuming that both feet = 100%. ** Pressure on the ground value calculated automatically for the forefoot of each taped foot assuming that a single treated foot = 100%.

## Data Availability

The data presented in this study are available on request from the corresponding author. The data are not publicly available due to the presence of sensitive data in each survey form.

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
