# Peer review of "The Influence of Short-Term Kinesiology Taping on Foot Anthropometry and Pain in Patients Suffering from Hallux Valgus"

_medicina, 2021, doi:10.3390/medicina57040313_

Round 1

Reviewer 1 Report

Thank you for inviting me to review this interesting paper. I think it is scientifically sound and worthy of publication with a few considerations, detailed below:

Major considerations:

While it is true that both the T1 and T2 trial demonstrated significantly reduced HV angles compared to pre-treatment, the HV angle goes up again in the later T2 trial compared to the earlier T1 trial. Thus, it is possible that the trend would continue to even higher HV angle after a longer period of time post-treatment. It is even conceivable that the effects disappear entirely with time. I think the authors should acknowledge this fact, and mention in the Discussion that longer durations of symptom reduction were not tested.

The statement of ethical considerations for studies on human subjects is vague and more information is needed. Did the authors submit an IRB proposal, or something comparable? Who reviewed the study proposal to determine that there were no ethical concerns? Please elaborate here.

Minor edits:

Line 9: improve should be improves

Lines 23-24: First sentence requires a citation.

Lines 33-36: I applaud the authors for mentioning the potential psychological effects of HV

Lines 42-44: The first two sentences of this paragraph contain slightly conflicting information. One states that the kinesiology tape is elastic while the other states that it is cotton. Perhaps the authors meant to define non-elastic taping in the second sentence? Please clarify.

Lines 65-66: The ethical approval process associated with the consent to participate in the study should be explained in more detail.

Line 141: I’d suggest expanding the first sentence or using a word other than “feelings”, such as experience or symptoms. As written, it sounds like it refers only to a patient’s emotional state rather than the experience of pain.

Lines 167-168: As mentioned above, it’s not entirely true that the effect was “maintained” in the T2 trial. The HV angle increased from T1 to T2, although it is still significantly lower in T2 than T0. I suggest rephrasing the word “maintained” here.

In the column heading for Table 1, “Parameter” is misspelled.

Author Response

Responses to Reviews

Dear Reviewers ,

We would like to thank you for positive review. Your comments were very useful and helped us to improve our manuscript.

Reviewer 1

Thank you for inviting me to review this interesting paper. I think it is scientifically sound and worthy of publication with a few considerations, detailed below:

Major considerations:

While it is true that both the T1 and T2 trial demonstrated significantly reduced HV angles compared to pre-treatment, the HV angle goes up again in the later T2 trial compared to the earlier T1 trial. Thus, it is possible that the trend would continue to even higher HV angle after a longer period of time post-treatment. It is even conceivable that the effects disappear entirely with time. I think the authors should acknowledge this fact, and mention in the Discussion that longer durations of symptom reduction were not tested.

The statement of ethical considerations for studies on human subjects is vague and more information is needed. Did the authors submit an IRB proposal, or something comparable? Who reviewed the study proposal to determine that there were no ethical concerns? Please elaborate here.

 Response: In the revised manuscript a sentences have been added to discussion:

The limitations of our study include a small research group, in the study we focused on the assessment of changes in the anthropometric parameters of the foot, hallux valgus angle, parameters of the static load of the foot, an additional aspect was also the assessment of subjective pain sensation in patients, which could be extended with the use of more complex forms and scales pain assessments for the foot. However, the promising results obtained in the studies show the need for further research on a larger group of patients and longer follow-up in this area to assess the long-term effects of Kinesiology Taping in patients with hallux valgus deformity.”

For the statement of ethical considerations for studies on human subjects we used standard forms approved by the Bioethical Committee of Medical University of Silesia (the signed forms were sent in the attachment to the editorial office). A detailed course of the experiment was presented to the Bioethical Committee of Medical University of Silesia, along with information on the type of tests and treatments performed on patients. The Bioethical Committee of Medical University of Silesia gave its consent to the experiment.

Information was provided in the manuscript: „The research project was consistent with the Declaration of Helsinki and was approved by the Bioethical Committee of Medical University of Silesia (approval number: KNW/0022/KB1/27/I/16). Before starting the study, all participants were clearly informed about the purpose and method of the research and their involvement in the entire process. All participants signed a consent form to participate in the experiment. In accordance with the recommendations and guidelines of Bioethical Committee of Medical University of Silesia, the form contains information on the objectives of the intended research and the manner of conducting it.”

Minor edits:

Line 9: improve should be improves

Response: In the revised manuscript the sentence was corrected:

„In this study we wanted to verify if Kinesiology Taping improves hallux valgus and affect the position as well as reducing pain.”

Lines 23-24: First sentence requires a citation.

Response: In the revised manuscript citation has been added:

„Hallux valgus (HV) is the most common osteoarticular deformation occurring within the forefoot [1,2].”

Lines 33-36: I applaud the authors for mentioning the potential psychological effects of HV

Response: Thank you, we also believe that the psychological aspect is a very important factor.

Lines 42-44: The first two sentences of this paragraph contain slightly conflicting information. One states that the kinesiology tape is elastic while the other states that it is cotton. Perhaps the authors meant to define non-elastic taping in the second sentence? Please clarify.

Response: Kinesiology Tape are types of elastic taping which stretch up to 140% of their original length. They are composed of cotton, elastic filaments, and highly durable adhesive glue that is both latex-free and waterproof, which provides tactile and mechanical stimulation.

In the revised manuscript a sentence has been added:

„Kinesiology Taping is a therapeutic method that uses the mechanical effect of a piece of elastic adhesive tape on the musculoskeletal system [14]. Kinesiology Taping uses cotton with elastic filaments tapes covered with acrylic adhesive on one side.”

Lines 65-66: The ethical approval process associated with the consent to participate in the study should be explained in more detail.

Response: In the revised manuscript these sentences have been added:

„The research project was consistent with the Declaration of Helsinki and was approved by the Bioethical Committee of Medical University of Silesia (approval number: KNW/0022/KB1/27/I/16). Before starting the study, all participants were clearly informed about the purpose and method of the research and their involvement in the entire process. All participants signed a consent form to participate in the experiment. In accordance with the recommendations and guidelines of Bioethical Committee of Medical University of Silesia, the form contains information on the objectives of the intended research and the manner of conducting it.”

Line 141: I’d suggest expanding the first sentence or using a word other than “feelings”, such as experience or symptoms. As written, it sounds like it refers only to a patient’s emotional state rather than the experience of pain.

Response: In the revised manuscript the sentence was corrected:

„Taping has a positive effect on the patients' subjective symptoms.”

Lines 167-168: As mentioned above, it’s not entirely true that the effect was “maintained” in the T2 trial. The HV angle increased from T1 to T2, although it is still significantly lower in T2 than T0. I suggest rephrasing the word “maintained” here.

Response: In the revised manuscript the sentence was corrected:

Taping affected the length of the foot (p <0.001) (Table 1). After the application of the taping, the length of the foot increased significantly (pT0vsT1 <0.01) and after one month of taping the foot length was still greater than before the KT application (pT0vsT2 <0.05).”

In the column heading for Table 1, “Parameter” is misspelled.

Response: In the revised manuscript the word „Parameter” in Table 1 was corrected.

Reviewer 2 Report

Line 61: what is the hypothesis of your study

Line 62: add how you recruited your subjects, also add for your measurements the precision and the CV of the measurements

Line 106 ff: were the tapes administered by the same person?

Lines 136-139: move this section to the beginning of the method section

Line 189: start the discussion with your main findings and compare with your hypothesis/hypotheses to discuss whether you could confirm or not your hypothesis/hypotheses

Line 247: add the strength, the weakness, the limitations and the implications for future research

Author Response

Responses to Reviews

Dear Reviewers ,

We would like to thank you for positive review. Your comments were very useful and helped us to improve our manuscript.

Reviewer 2

Line 61: what is the hypothesis of your study

Response: The hypothesis of our study is that Kinesiology Taping applicated on foot can be able to correct the big toe position and reduce pain symptoms which may be an alternative method for surgical treatment of this deformity in patients who cannot undergo surgery.

Line 62: add how you recruited your subjects, also add for your measurements the precision and the CV of the measurements

Response: In the revised manuscript the sentence was corrected:

„Twenty-three people aged 26.3 to 78.4 years, mean ± standard deviation: 55.8 ± 16.3 years,  were involved in the study. They were all patients with symptomatic HV who attended the Healthy Foot Clinic, and wanted to undergo non-operative treatment and expressed their willingness to participate in the study.”

Line 106 ff: were the tapes administered by the same person?

Response: In the revised manuscript the sentence was corrected:

„which is why all patients were taped with Kinesiology Taping application by the same physiotherapist every 2-3 days”

Lines 136-139: move this section to the beginning of the method Section

Response: In the revised manuscript, the above paragraph was moved to the beginning of the method section and extended at the suggestion of other reviewers.

Line 189: start the discussion with your main findings and compare with your hypothesis/hypotheses to discuss whether you could confirm or not your hypothesis/hypotheses

Response: In the revised manuscript these sentences have been added:

„Considering that not every patient can or wants to undergo surgery, effective methods of conservative treatment should be sought to help such patients. Promising results obtained in our study show that using Kinesiology Taping might be the method of conservative treatment”

Line 247: add the strength, the weakness, the limitations and the implications for future research

Response: In the revised manuscript these sentences have been added in discussion section:

 „The limitations of our study include a small research group, in the study we focused on the assessment of changes in the anthropometric parameters of the foot, hallux valgus angle, parameters of the static load of the foot, an additional aspect was also the assessment of subjective pain sensation in patients, which could be extended with the use of more complex forms and scales pain assessments for the foot. However, the promising results obtained in the studies show the need for further research on a larger group of patients and longer follow-up in this area to assess the long-term effects of Kinesiology Taping in patients with hallux valgus deformity.”

Reviewer 3 Report

This is an interesting study, which has been conducted and presented well by the authors. It provides an apparently effective alternate treatment option in those patients who are not suitable for surgery. My only question to authors is that what happened to these patients after one month (T2 stage)? How long were they followed up beyond one month stage, and were any measurements done at a later stage? as the effect of taping for one months would be temporary and the deformity is likely to return due to soft tissue contractures etc.

Overall, the study has merits to be published despite small numbers and shorter follow up, in view of the fact that this treatment option is not curative and is only for symptomatic benefit in a small minority of patients.

Author Response

Responses to Reviews

Dear Reviewers ,

We would like to thank you for positive review. Your comments were very useful and helped us to improve our manuscript.

Reviewer 

This is an interesting study, which has been conducted and presented well by the authors. It provides an apparently effective alternate treatment option in those patients who are not suitable for surgery. My only question to authors is that what happened to these patients after one month (T2 stage)? How long were they followed up beyond one month stage, and were any measurements done at a later stage? as the effect of taping for one months would be temporary and the deformity is likely to return due to soft tissue contractures etc.

Overall, the study has merits to be published despite small numbers and shorter follow up, in view of the fact that this treatment option is not curative and is only for symptomatic benefit in a small minority of patients.

Response: After the experiment, the patients were taught to apply the tapes on their own and were recommended individual additional therapeutic elements, e.g. mobilization of the MTP1 joint, Cyriax transverse friction procedure on lateral ligament, relaxation of hallucis adductor muscle. Unfortunately, further observations were disrupted by the pandemic.

Reviewer 4 Report

Paper is of interest to the scietific community. Some clarifications should be made before accept it for publication:

1.- Change "Hallux Valgus" into "Hallaux Abductus Valgus". The last one is the accepted term for scietific community. 

2.- Use keybords included in MeSH ("hallux valgus" is correct).

3.- Renove "dynamic tape" (line 42). It is a differnte kind of taping that used in this paper.

4.-About sample: authors must clarify about patients clinical situation. Have the patients been under pharmacological treatment? Have the patients been using foot orthoses?

5.- About taping treatment: authors state patiens have been using taping along one month. Please clarify how many times taping has been replaced.

6.- Explain how HV angle has been measured without using a Rx.

Author Response

Responses to Reviews

Dear Reviewers ,

We would like to thank you for positive review. Your comments were very useful and helped us to improve our manuscript.

Reviewer

Paper is of interest to the scietific community. Some clarifications should be made before accept it for publication:

1.- Change "Hallux Valgus" into "Hallaux Abductus Valgus". The last one is the accepted term for scietific community. 

Response: The term Hallux Abductus Valgus appears from time to time, most authors write "Hallux valgus" - PubMed HAV: HV = 39 works out of 2728. At the request of the editors, the authors agree to change the name hallux valgus to Hallux Abductus Valgus, but suggest to stay with the name hallux valgus.

2.- Use keybords included in MeSH ("hallux valgus" is correct).

Response: Unfortunately we did not understand the reviewer's intention. We are asking for a broader explanation.

3.- Remove "dynamic tape" (line 42). It is a differnte kind of taping that used in this paper.

Response: "dynamic tape"  removed at the reviewer's suggestion

4.-About sample: authors must clarify about patients clinical situation. Have the patients been under pharmacological treatment? Have the patients been using foot orthoses?

Response: During the experiment, no other methods of treatment were used by patients, including orthopedic insoles, orthoses, separators, pharmacological treatments.

5.- About taping treatment: authors state patients have been using taping along one month. Please clarify how many times taping has been replaced.

Response: About 10-14 applications were applied to the patients - it depends on how long the skin was sweating and how long the tape remained on the skin

Information was provided in the manuscript „ …all patients were taped with Kinesiology Taping application by the same physiotherapist every 2-3 days in clinic in a period of about a month”

6.- Explain how HV angle has been measured without using a Rx.

Response: The HV angle was calculated automatically by a computer program on the basis of a 3D scan of the foot.